# Humanlike spontaneous motion coordination of robotic fingers through spatial multi-input spike signal multiplexing

Dong Gue Roe[1,7], Dong Hae Ho[2,7], Yoon Young Choi[3], Young Jin Choi[2], Seongchan Kim[4], Sae Byeok Jo [5], Moon Sung Kang [6], Jong-Hyun Ahn [1] & Jeong Ho Cho [2] ✉

With advances in robotic technology, the complexity of control of robot has been increasing owing to fundamental signal bottlenecks and limited expressible logic state of the von Neumann architecture. Here, we demonstrate coordinated movement by a fully parallel-processable synaptic array with reduced control complexity. The synaptic array was fabricated by connecting eight ion-gel-based synaptic transistors to an ion gel dielectric. Parallel signal processing and multi-actuation control could be achieved by modulating the ionic movement. Through the integration of the synaptic array and a robotic hand, coordinated movement of the fingers was achieved with reduced control complexity by exploiting the advantages of parallel multiplexing and analog logic. The proposed synaptic control system provides considerable scope for the advancement of robotic control systems.

In recent decades, advances in robotic control systems have enabled more sophisticated and delicate robot movement control[1-5]. However, high-level robotic control has inevitably increased the complexity of the control system. Although efforts have been made to develop efficient control systems, current CMOS-based robotic control systems have inherent limitations— the von Neumann bottleneck and the binary logic structure— resulting in signal delay and poor chip integration[6-12]. The advent of synaptic transistors provided a breakthrough in addressing signal delay and chip integration issues[13-17]; these devices are capable of parallel computation and analog signal processing, similar to the human nervous system[18-22]. Among the various types of synaptic transistors[6,12,23-27], ion-based electrochemical synaptic transistors are attracting attention for biomimetic applications since their working mechanism is similar to that of human synapses. Human synapses transmit biological signals by releasing neurotransmitters from presynaptic neurons; the neurotransmitters pass through the synaptic cleft and reach postsynaptic neurons[18,20-22,28-30]. In the case of an electrochemical synaptic transistor, signal transmission occurs through the electrical-input-signal-induced penetration of a semiconducting channel by the ionic species. Ionic movement in an electrochemical synaptic transistor depends on the transmission distance. This unique property of synaptic transistors facilitates the control of multiple devices with a single input signal through the selection of an appropriate signal transmission distance and the integration of multiple signals from each device into a unified signal[31,32]. The multi-control and signal processing capabilities of ion gel reduce computational efforts when synaptic transistors are used in robotic systems, since they allow parallel control under a reduced input signals and efficient distance-based signal multiplexing. Owing to their analog processing capability, synaptic transistors also help enhance robotic control performance by

[1]School of Electrical and Electronic Engineering, Yonsei University, Seoul 03722, Republic of Korea. [2]Department of Chemical and Biomolecular Engineering, Yonsei University, Seoul 03722, Republic of Korea. [3]Department of Mechanical Science and Engineering, University of Illinois at Urbana–Champaign, Urbana, IL 61801, USA. [4]SKKU Advanced Institute of Nanotechnology (SAINT), Sungkyunkwan University, Suwon 16419, Republic of Korea. [5]School of Chemical Engineering, Sungkyunkwan University, Suwon 16419, Republic of Korea. [6]Department of Chemical and Biomolecular Engineering, Institute of Emergent Materials, Sogang University, Seoul 04107, Republic of Korea. [7]These authors contributed equally: Dong Gue Roe, Dong Hae Ho. ✉e-mail: jhcho94@yonsei.ac.kr

allowing digital-to-analog circuits (DACs)—an essential circuit component of conventional CMOS-based control systems—to be omitted and thereby reducing the circuit complexity of the control system. The combination of these core characteristics of synaptic transistors is helpful for realizing complex actuation such as the coordinated movement of a human finger[33–35] with simplified circuits and fewer computational requirements.

In this study, we developed a fully parallel-processable control system capable of signal processing, by promoting free ionic movement in the dielectric of an electrochemical synaptic transistor. The system was constructed by connecting an ion-gel-based parallel-processable synaptic array (PPSA) to a coordinated robotic hand (CRH). These two control system components were fabricated using eight organic artificial synaptic transistors (OASTs) that shared a common ion gel dielectric and assembling three independent NiTi shape memory alloy fiber-based robotic fingers in a 3D-printed body, respectively. First, it was verified that three OASTs could operate simultaneously with a single input and that their synaptic output could be adjusted by changing the distance between the input gate and the transistor. Subsequently, the bending angle of robotic finger for different synaptic outputs was evaluated. Finally, coordinated robotic actuation of grabbing an object with curvature and a complex design was successfully performed for an optimized input gate-transistor distance by using a circuit architecture with reduced complexity. The results of this study are expected to contribute to the improvement of the control efficiency of humanoids and animatronics, for which complex calculations are a requisite for robotic operation.

## Results

### Parallel-processable robotic control system capable of ion-based signal multiplexing

Figure 1a shows a comparison of the proposed synapse-based control system and a CMOS-based conventional control system. The former has two advantages over the latter: spatial parallel signal multiplexing capability and the absence of a DAC. Generally, in CMOS electronics, a transistor can process a signal only serially, whereas a synapse-based control system is capable of multiplexing several signals (i.e., parallel signal processing) owing to free ionic movement in the ion gel dielectric. Ion-based signal multiplexing allows signals to be processed without the use of complex processors while reducing circuit components required for the input selector, which may cause signal delays because of the von Neumann bottleneck. The other merit of the proposed system—the absence of a DAC—results from the inherent characteristics of synaptic transistors: they can process analog signals and directly transmit them to actuators without any computational process being involved. Consequently, unlike a digital-signal-based CMOS control system that requires a DAC for signal type conversion, the proposed system does not require a DAC. Both these advantages of the synapse-based control system simplify the circuit architecture and contribute to efficient signal processing, which plays a crucial role in controlling complex actuation, especially for realizing coordinated human motion in the robotic system. With conventional control system, mimicking human motion generated by a combination of adjacent muscle actuation requires enormous computational resources, to control several actuators simultaneously. By contrast, the use of a synapse-based control system considerably reduces the computa-

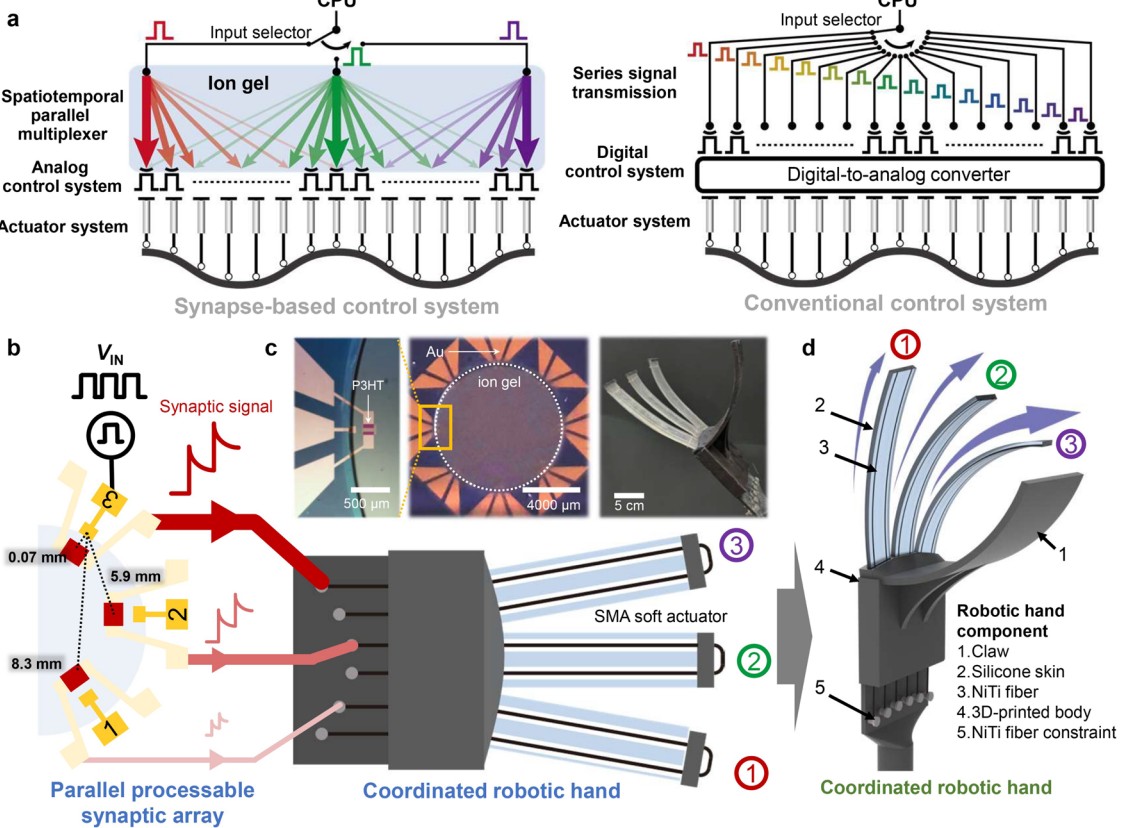

**Fig. 1 | Parallel processable robotic control system capable of ion-based signal multiplexing. a** Schematic comparison between a synapse-based control system and a conventional control system. **b** Schematic of CRH control through the PPSA. **c** Photographs of an OAST, the PPSA, and the CRH. **d** Three-dimensional diagram of the coordinated actuation of the CRH and its components.

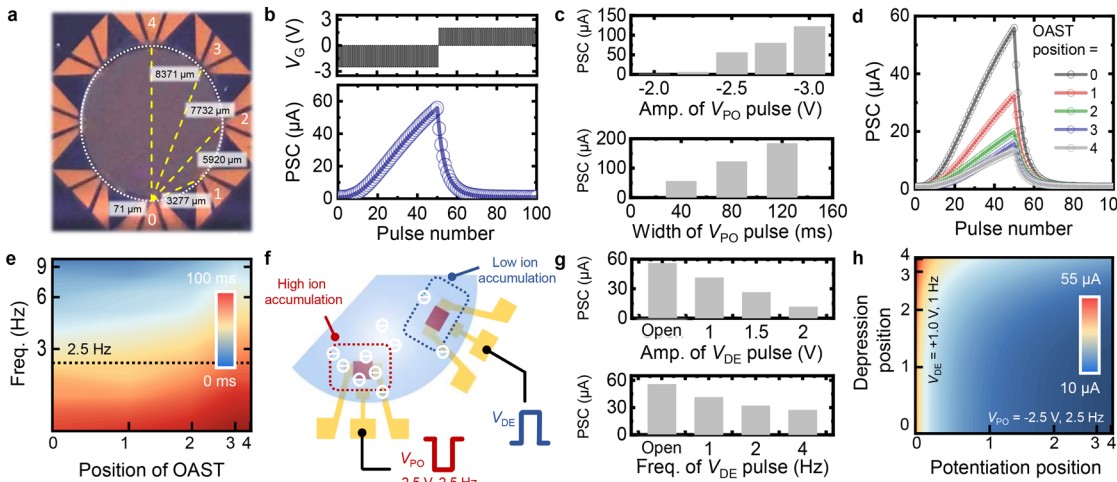

**Fig. 2 | Characterization of the PPSA. a** A photograph of the PPSA, with its dimensions shown. **b** PSC change of an OAST under consecutive 50 $V_{PO}$ pulses and 50 $V_{DE}$ pulses ($V_{PO}$ = 2.5 V, 40 ms and $V_{DE}$ = 2 V, 40 ms). **c** PSC change for various $V_{PO}$ conditions (amplitude of $V_{PO}$: −2.0 to −3.0 V; width of $V_{PO}$: 40–120 ms). **d** PSC change with increasing distance between the applied gate and an OAST.

**e** Calculated signal delay for different OAST positions and frequencies. **f** Schematic of the signal multiplexing mechanism when $V_{PO}$ and $V_{DE}$ are applied simultaneously. **g** PSC change for different $V_{DE}$ values (amplitude of $V_{DE}$: open to 2.0 V; frequency of $V_{DE}$: open to 4 Hz). **h** Change in PSC with the application position of $V_{PO}$ and $V_{DE}$.

tional effort since the surrounding actuators are automatically manipulated on the basis of their distance from the main actuator.

Finger movement is among the human motions that are highly dynamic and difficult to describe. Hence, a synapse-based control system is advantageous for realizing artificial finger movements. Figure 1b shows a schematic of CRH control. The OAST in the PPSA is connected to the robotic fingers of the CRH. Among the three gate electrodes of OASTs, gate 3 serves as the input terminal and a voltage pulse is applied to it. In the OAST configuration, the distance from gate 3 to each OAST channel is determined from the geometry: the distance increases as the gate position number decreases from 3 to 1. Since an increase in the distance results in a larger signal delay, the synaptic signal by the input voltage pulse was the weakest at OAST 1 and the strongest at OAST 3. Three synaptic signals with different strengths were transmitted to each robotic finger for the bending actuation of the CRH. The degree of bending depended on (and was thereby controllable through) the amplitude and width of synaptic signals as well as the distance of signal transmission. Figure 1c and Supplementary Fig. 1 show conceptual and photographic images of the PPSA and CRH. The PPSA was fabricated by connecting eight OASTs that share a common ion gel dielectric (see Supplementary Fig. 2 for the detailed fabrication process), and the CRH consisted of a 3D-printed claw and a body connected to three robotic fingers. The robotic fingers were constructed by embedding NiTi shape memory alloy fiber into PDMS matrix (see Supplementary Fig. 3 for the detailed fabrication process of the CRH).

**Characterization of PPSA**
Figure 2a shows an optical image of the PPSA, which comprised eight identical OASTs arranged in a circle. Notably, the eight OASTs shared a single ion-gel dielectric layer and were therefore ionically connected. A potentiation pulse applied to one OAST could hence be transmitted to the channels of the other OASTs (of course, this transmission would depend on the distance between the gate electrode and the OAST channel). Furthermore, multiple potentiation pulses applied to the gates could be multiplexed via the shared ion-gel dielectric to generate integrated signals for the OASTs. Before analyzing the response of the PPSA to the multiplexed potentiation input, we first examined the characteristics of an OAST when a single potentiation input was provided. The operation of a single OAST can be explained on the basis of the motion of ionic species in the ion gel. When a negative potentiation

voltage ($V_{PO}$) pulse is applied to the gate electrode, anions (i.e., [PF$_6$]) in the ion gel are driven into the free volume of the poly(3-hexylthiophene-2,5-diyl) (P3HT) channel. The [PF$_6$] anions penetrating the P3HT channel induce hole accumulation in the channel and thereby cause the channel conductance to increase. On the other hand, when a positive depression voltage ($V_{DE}$) pulse is applied to the gate electrode, [PF$_6$] anions that have penetrated the P3HT channel are extracted from the P3HT channel, which resets the conductance of the OAST. Figure 2b shows the long-term potentiation/depression (LTP/D) characteristic of an OAST. A linear update in the post synaptic current (PSC) was observed because of the potentiation pulse that caused the accumulation of [PF$_6$] anions in the P3HT channel. On the other hand, when the depression pulse was applied to the input gate, the as-accumulated anions were gradually extracted from the P3HT channel, leading to a decreased in the PSC. The synaptic performance of the OAST was measured under different input pulse conditions (width and amplitude), as shown in Fig. 2c and Supplementary Fig. 4. With an increase in the pulse amplitude and pulse width, a higher PSC was obtained because of larger accumulation of anions in the P3HT channel. Also, repeatability and environmental stability of the OAST was evaluated by applying 8,000 LTP/D pulses and changing temperature and humidity, respectively (Supplementary Fig. 5).

Next, we examined the synaptic characteristics of a single OAST as a function of the distance between the gate electrode and the channel. Specifically, the PSC of an OAST channel was measured by applying a potentiation signal to the gate electrodes at different positions (Positions 0 to 4 in Fig. 2a). The PSC decreased from 55.8 μA to 13.2 μA as the gate varied from Position 0 to Position 4 (Fig. 2d). Electrochemical impedance spectroscopy (EIS) measurement was conducted to investigate the distance effect. An AC potential (±25 mV) was applied to the gates at different positions, and the values of Z′ and Z″ were plotted (Supplementary Fig. 6). Figure 2e shows the estimated RC delay of OASTs acquired at different AC frequencies through potentiation signals applied at different positions. Here, the RC delay is inherited delay that roots from the resistive and capacitive characteristics of the ion gel connected in series[32]. The delay constants were estimated by multiplying the resistance and capacitance values obtained from electrochemical impedance spectroscopy (EIS). Clearly, a larger signal delay was generated by longer distance between gate and channel; for the AC frequency of 2.5 Hz, which was used for the PSC measurement, the delay time increased from 13 to 33 ms as the OAST position

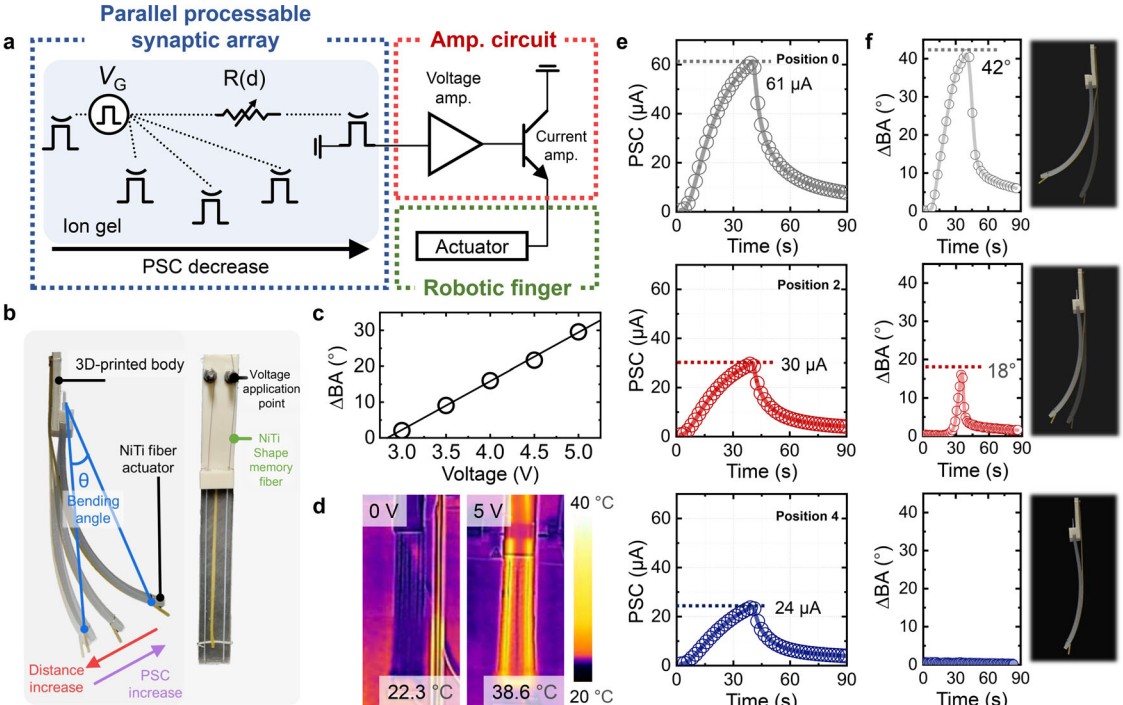

**Fig. 3 | Characterization of robotic finger control with the PPSA. a** Circuit diagram of the robotic finger actuation system. **b** Photographs of the fabricated robotic hand and its components. **c** Change in the BA of a robotic finger with increasing applied voltage on the robotic finger (3.0–5.0 V). **d** Thermal images of a robotic finger when the applied voltage on the robotic finger is increased. **e** PSC measured in an OAST at various positions. **f** Change in the BA of a robotic finger at various positions and the finger's photographs (the translucent images indicate the original position of the robotic finger).

changed from 0 to 4. The larger time delay indicates slower signal transmission of the PPSA, which eventually leads to a decreased PSC. Among various conditions we tested (see Supplementary Fig. 7), the largest difference in the PSC for inputs at Positions 0 and 4 was obtained using a 2.5 Hz $V_{PO}$ pulse with a magnitude −2.5 V and a width of 40 ms. This $V_{PO}$ condition was used for the following experiment.

The spatial multiplexing performance of the PPSA was investigated by applying two independent inputs to two gate electrodes of the PPSA separately. Figure 2f shows a schematic of ion accumulation when two different pulses were applied to two different gate electrodes of the PPSA. When $V_{PO}$ (facilitates anion penetration into P3HT) and $V_{DE}$ (facilitates anion extraction from P3HT) were applied to two different gate electrodes, the PSC resulting from the multiplexing of the two pulses within the shared ion-gel layer changed depending on both the amplitude and the width of the respective input pulses. Figure 2g summarizes the effect of multiplexing on the PSC for the OAST at Position 0. PSCs were acquired by varying the amplitude and frequency of the $V_{DE}$ pulse applied to Position 1, while maintaining the $V_{PO}$ pulse applied to Position 0 at a fixed condition (amplitude of 2.5 V at 2.5 Hz) (see LTP/D characteristics in Supplementary Fig. 8). Increasing the pulse amplitude and frequency of the $V_{DE}$ pulse led to the extraction of anions from the P3HT channel, and accordingly, a lower PSC was obtained. The PSC response under constructive potentiation and complex input situations was also investigated. Potentiating the PPSA with multiple inputs constructively increased the PSC; a PSC larger than that obtained using a single input at Position 0 was observed (Supplementary Fig. 9). Furthermore, various PSC levels could be achieved through different combinations of input signals applied at different Positions (Supplementary Fig. 10). Figure 2h maps the effect of potentiation and the depression position on the PSC recorded at Position 0; the $V_{PO}$ pulse had an amplitude of −2.5 V and a frequency of 2.5 Hz, and the $V_{DE}$ pulse was provided at an amplitude of 1.0 V at 1.0 Hz (Supplementary Fig. 11). Overall, the amplitude of the multiplexed signal could be modulated by both input voltage pulse ($V_{PO}$ and $V_{DE}$)

and distance between the measured OAST and the gate electrode to which a voltage pulse was applied.

## Characterization of robotic finger control with PPSA

Figure 3a shows a schematic circuit diagram of the robotic finger actuation system. The input signals were parallelly multiplexed by the PPSA and then transmitted to the amplifier circuit that amplified the synaptic output for robotic finger actuation. A photograph of a robotic finger is shown in Fig. 3b. The robotic finger comprised a 3D-printed body and PDMS matrix in which NiTi shape memory alloy fiber was embedded. When a PSC was applied to the NiTi fibers, a phase transformation from martensite to austenite occurred because of Joule heating, which caused structural deformation and shortened the fiber length. The shortened fiber induced compressive (shallowly embedded NiTi fiber side) and tensile (the opposite side) stresses to bend the robotic finger. The actuation of the robotic finger (i.e., robotic finger bending) could be quantified in terms of the bending angle (BA), defined as the angular displacement between the free-end and the fixed-end, which served as the central axis. Figure 3c shows the relationship between the BA and the voltage applied to the robotic finger. A linear increase in the BA was observed above a threshold voltage of 3 V. Figure 3d and Supplementary Fig. 12 show the thermal image of the robotic finger as a function of the applied voltage. The temperature of the NiTi fiber increased with the applied voltage, and consequently, a larger BA was obtained. The repeatability of the robotic finger was tested by performing 15 consecutive actuation tests (Supplementary Fig. 13).

The analog control of robotic finger on the basis of the signal transmission distance of the PPSA was tested by connecting the PPSA to a robotic finger. Figure 3e, f and Supplementary Fig. 14 show the PSC of the OAST and the BA of the robotic finger for different input gate positions. The robotic fingers were connected to five OASTs of the PPSA (0, 1, 2, 3, and 4), and 50 LTP (−2.5 V, 2.5 Hz) and 50 LTD (1 V, 2.5 Hz) pulses were applied consecutively at the gate position 0. As the

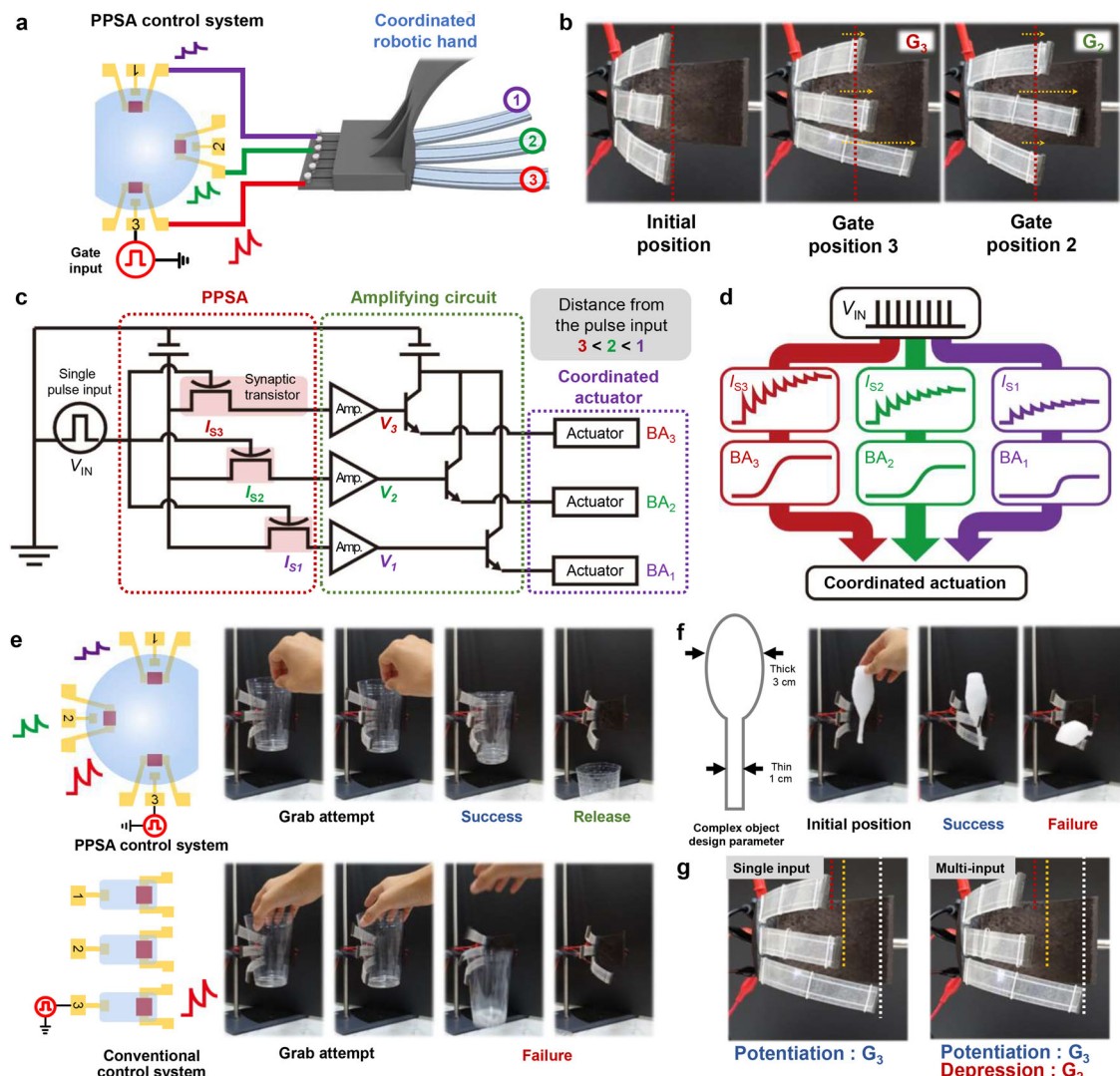

**Fig. 4 | CRH control with PPSA. a** Schematic illustration of connection between PPSA control system and CRH. **b** Photographic image of coordinated actuation under varying applied input position. **c** Circuit diagram of coordinated actuation system. **d** Schematic illustration of signal flow of coordinated actuation. **e** Comparison of cup grabbing performance between the PPSA control system and the conventional control system. **f** Comparison of complex object grabbing performance between the PPSA control system and the conventional control system. **g** Photographic image of showing BA change under applying multi-input to PPSA control system.

distance between the OAST and the input gate increased, the PSC decreased from 61 to 24 µA because of slower signal transmission through the ion gel, and the BA decreased from 42° to 0°. This indicates that the PSC magnitude can be adjusted by choosing an appropriate OAST, and thereby, confirming a controllability of distance-based BA control system. Note that because the PSC was adjustable in analog states, a digital-to-analog converter was not required during signal transmission.

### Coordinated robotic hand control with PPSA

Finally, the PPSA, amplifying circuits, and three robotic fingers were integrated into a CRH capable of parallel signal multiplexing and analog control (Fig. 4a). Three of the PPSA's eight OASTs were selected, and they were connected to three robotic fingers via voltage and current amplifiers. Figure 4b shows the coordinated actuation of the CRH with different gate inputs. When input pulses were applied to gate position 3, the most significant BA change was observed for robotic finger 3, whose synapse (OAST 3) was the closest to the input gate. The BAs of robotic fingers 1 and 2 showed smaller changes owing to the larger distance between the OASTs and

the input gate. Similarly, when input pulses were applied to gate position 2, the largest BA change occurred at robotic finger 2. Figure 4c, d shows the circuit diagram and the signal flow of coordinated actuation, respectively. When input voltage ($V_{IN}$) pulses were applied to the gate electrode, signals were parallelly transmitted to the three OASTs through the ion gel of the PPSA. The amplitude of the $V_{IN}$ pulse was controlled by the distance between the input gate electrode and the OAST, and the transmitted $V_{IN}$ pulses simultaneously generated different PSCs according to the gate-OAST distance. The PSCs were then amplified by a series of voltage and current amplifiers and delivered to the actuator. Consequently, coordinated actuation was induced with three different BAs that depended on the distance between the input gate and each OAST. Figure 4e compares the grabbing performance of a CRH between the PPSA-based control system and a conventional control system. Both systems were used for holding a cup containing 10 mL of water, and $V_{IN}$ was applied to gate position 3. In the case of the PPSA-based control system, the three robotic fingers successfully grabbed the cup firmly along the contour of cup through coordinated movement induced by the LTP pulses and released the cup when $V_{IN}$ was decreased by the

consecutive LTD pulses (Supplementary Video 1). However, in the latter case, the grabbing action driven by only one signal transmission failed because a single $V_{IN}$ could bend only the third robotic finger (Supplementary Video 2). The grabbing of an object with a more complex shape is shown in Fig. 4f. Again, the PPSA control system successfully grabbed the spoon-shaped object with a single $V_{IN}$, while the conventional system failed. The use of a multi-input signal that is parallelly multiplexed can enhance the grabbing action through more delicate robotic finger control. Figure 4g shows the CRH under multi-input control. The $V_{PO}$ and $V_{DE}$ pulses were applied to gate positions 3 and 2, respectively. Since the depression pulse hindered the PSC increase in OASTs 1 and 2, the BA change of robotic fingers 1 and 2 decreased compared with the case where a single input was provided. These results demonstrate that our system is capable of humanlike coordinated actuation of a robotic hand through parallel signal processing with reduced circuit components.

## Discussion

The PPSA-based synaptic control system is capable of coordinated actuation through parallel multiplexing and analog control resulting from the characteristics of signal multiplexing in the shared ion gel of PPSA. The control system comprised the PPSA and CRH. The robotic fingers of the CRH were controlled simultaneously by a single input that could be modulated by the signal transmission distance in the PPSA. The coordinated actuation performance of the control system was verified for grabbing action by the CRH by using a cup and a complex-shaped object. The proposed PPSA-based synaptic control system serves as an efficient signal processing platform and provides a breakthrough for robotic control systems involving massive amounts of computation.

## Methods

### Device fabrication

**Parallel processable synaptic array.** The processing solvents and precursors were purchased from Sigma-Aldrich. Poly(3-hexylthiophene) (P3HT) solution was prepared by dissolving P3HT in chloroform at a concentration of 7 mg/ml and stirring the resulting solution for 3 h at 50 °C under ambient conditions. The ion gel ink was prepared by mixing poly(ethylene glycol)diacrylate (PEGDA) monomer, 2-hydroxy-2-methylpropiophenone (HOMPP) initiator, and 1-butyl-3-methylimidazolium hexafluorophosphate ([BMIM][PF$_6$]) ionic liquid in a weight ratio of 2:1:12. The synaptic array was fabricated on a SiO$_2$/Si substrate. Cr/Au (3/17 nm) was deposited on the substrate through thermal evaporation and gate, drain, and source electrodes were patterned through photolithography (AZ 5214E) and chemical etching (Chrome etchant CE-905N, Gold etchant TFA). A P3HT channel was then deposited through spin-coating (2000 rpm for 45 s), and it was patterned through photolithography and reactive ion etching. Finally, the prepared ion gel ink was drop-cast onto the substrate and patterned under UV irradiation (100 mW cm$^{-2}$ at 365 nm for 8 s) through a photomask.

**Coordinated robotic hand.** The body of the robotic hand was prepared using a 3D printer (ANYCUBIC, Photon Mono X). A mold with dimensions of 100 mm × 20 mm × 3 mm (length × width × height) was designed and prepared using a 3D printer for producing the PDMS robotic finger body. The path of NiTi shape memory alloy fiber was set by placing two polytetrafluroethylene (PTFE, Misumi, TUBF26-10, diameter = 0.92 mm) tubes in the mold. Subsequently, the uncured PDMS (Sylgard 184) was poured into the mold coupled with the PTFE tube and placed in a vacuum for 30 min to remove air bubbles in the PDMS. The PDMS was then cured at 80 °C. The mold was removed, and the NiTi fiber was threaded through the PTFE tube to complete the robotic finger. Finally, three robotic fingers were integrated into the body of the robotic hand.

### Measurements

The electrical properties of the all devices were measured with a Keithley 4200A-SCS. The bending of the robotic finger was video recorded on robotic hand's side, and BAs were estimated by analyzing the recorded videos using the ImageJ PhotoBend plugin. Electrochemical impedance spectroscopy was performed with an AMETEK Princeton Applied Research VersaSTAT4 potentiostat. Thermal images were obtained using FLIR E6-XT.

## Data availability

The data for the plots presented in this paper and other findings of this study are available from the corresponding author upon reasonable request.

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

## Acknowledgements

J.H.C. was supported by the Basic Science Program (NRF-2020R1A2C2007819) through the National Research Foundation (NRF) of Korea funded by the Ministry of Science and ICT, Korea, Creative Materials Discovery Program (NRF-2019M3D1A1078299) through the National Research Foundation (NRF) of Korea funded by the Ministry of Science and ICT, Korea, and Korea Medical Device Development Fund grant funded by the Korea government (the Ministry of Science and ICT) (Project Number: KMDF202012B02-02). J.H.C. was partially supported by the Yonsei Signature Research Cluster Program of 2021 (2021-22-0004).

## Author contributions

J.H.C. initiated and supervised all the research. D.G.R. and D.H.H. carried out and designed most of the experimental work and data analysis. Y.Y.C. assisted in the writing of the manuscript. Y.J.C. and S.K. assisted in the conceptualization and validation of research. M.S.K., J.H.A., and S.B.J. assisted in the data analysis. All authors discussed the results and contributed to the writing of the manuscript.

## Competing interests

The authors declare no competing interests.
