## [Peer Review File · Nature Communications]

REVIEWER COMMENTS

Reviewer #2 (Remarks to the Author):

Great to see the proposed humanlike multi-input system for the control of robotic fingers. This is an interesting topic and the authors have provided reliable data to support it. It would be great if authors could address these below before consideration of publication.

1. Line 141: Please add more detail about 'RC delay'? Also, how do you demonstrate the 'calculate signal' in line 370?
2. Line 154-159: The description is about the effect of amplitude and width on PSC. However, Figure 2g is about the effect of Amplitude and Frequency on PSC. It needs to be consistent.
3. Line 166: The description of "It was found that the multiplexed effect depended more strongly on V_{de} than V_{po} " is not that clear from Supplementary Fig. 10. Please add more detailed explanation here.
4. Line 190: "..., the PSC decreased from 61 μA to 24 μA ..." is this PSC mentioned here as the Maximum of PSC? If so, why there is a peak value? How did the author obtain this value? Authors need to add more experimental descriptions for Fig. 3e and 3f.
5. Line 198: The author chose three out of the eight OASTs to control the fingers of the robot. The question here is why the author did not make five fingers instead of three to mimic a human hand?
6. Please be careful when you use the word "spatiotemporal" because the input signals here are only about the effect from distance (space information). There is no effect of the time-related input signal that has been demonstrated in this manuscript.
7. Have authors considered the method of how to initiate the state of each finger after bending?

8. Authors also may consider cite more references in the context of electrochemical synaptic transistor's use for the robot manipulation in line 47-48, such as "A 3D-printed neuromorphic humanoid hand for grasping unknown objects", *Science*, Volume 25, Issue 4, 15 April 2022, 104119.

Reviewer #3 (Remarks to the Author):

The manuscript titled "Humanlike Spontaneous Motion Coordination of Robotic Fingers Through Spatiotemporal Multi-input Spike Signal Multiplexing" by Dong Gue Roe et al. exhibited a coordinated movement of the fingers was achieved with reduced control complexity by exploiting the advantages of parallel multiplexing and analog logic by the integration of the synaptic array and a robotic hand. In general, this work seems to be quite interesting and I support its publication but I have some comments.

1. The properties of ion gel dielectric can be influenced by external environment, such as humidity, temperature, thus the operating environment of robotic hand should introduce.
2. How about the repeatability of transistor and the bending angle of robotic finger?
3. How long the coordinated robotic actuation of grabbing an object with curvature can maintain?
4. In fig2d, there's some overlap between 3 and 4.

Reviewer: 1

Great to see the proposed humanlike multi-input system for the control of robotic fingers. This is an interesting topic and the authors have provided reliable data to support it. It would be great if authors could address these below before consideration of publication.

1. Line 141: Please add more detail about ‘RC delay’? Also, how do you demonstrate the ‘calculate signal’ in line 370?

→ Thank you for the detailed advice on our manuscript. The RC delay in the proposed PPSA is a parasitic signal delay caused by the resistive and capacitive characteristics of ion gel connected in series.¹ The resistance and capacitance of the ion gel could be obtained by using electrochemical impedance spectroscopy (EIS) with varying input frequency. These values were multiplied to estimate the RC delay constant. In the revised manuscript, we provide additional information on the estimation of the RC delay as follows:

Line 144: Here, the RC delay is inherited delay that roots from the resistive and capacitive characteristics of the ion gel connected in series.³² The delay constants were estimated by multiplying the resistance and capacitance values obtained from electrochemical impedance spectroscopy (EIS).

2. Line 154-159: The description is about the effect of amplitude and width on PSC. However, Figure 2g is about the effect of Amplitude and Frequency on PSC. It needs to be consistent.

→ Thank you for your comment. As the reviewer commented, using the term “pulse width” and “frequency” may have confused the reader/reviewer. To avoid the confusion, we revised the manuscript as follows:

Line 160: PSCs were acquired by varying the amplitude and frequency of the V_{DE} pulse applied to Position 1, while maintaining the V_{PO} pulse applied to Position 0 at a fixed condition (amplitude of 2.5 V at 2.5 Hz) (see LTP/D characteristics in **Supplementary Fig. 8**). Increasing the pulse amplitude and frequency of the V_{DE} pulse led to the extraction of anions from the P3HT channel, and accordingly, a lower PSC was obtained.

3. Line 166: The description of “It was found that the multiplexed effect depended more strongly on V_{de} than V_{po} ” is not that clear from Supplementary Fig. 10. Please add more detailed explanation here.

→ Thank you for the comment. We claimed that “the multiplexed effect depended more strongly on V_{DE} than V_{PO} ” because V_{DE} led significant current decreasing effect despite of the smaller amplitude and slower frequency compared to those of V_{PO} . However, as the reviewer noted, that information does not seem to be drawn clearly from Supplementary Figure 10. To enhance the understanding of the readers, we removed the statement in the revised manuscript.

4. Line 190: “..., the PSC decreased from 61 μ A to 24 μ A...” is this PSC mentioned here as the Maximum of PSC? If so, why there is a peak value? How did the author obtain this value? Authors need to add more experimental descriptions for Fig.3e and 3f.

→ Thank you for the valuable comment. The PSC current and the bending angle depicted on Fig.3e and 3f were obtained by applying 50 LTP pulses and 50 LTD pulses consecutively. The peak value of the PSC and the bending angle was obtained at the end of the 50 LTP pulses. To clarify this, we added experimental detailed as follows:

Line 193: The robotic fingers were connected to five OASTs of the PPSA (0, 1, 2, 3, and 4), and 50 LTP (-2.5 V, 2.5 Hz) and 50 LTD (1 V, 2.5 Hz) pulses were applied consecutively at the gate position 0.

5. Line 198: The author chose three out of the eight OASTs to control the fingers of the robot. The question here is why the author did not make five fingers instead of three to mimic a human hand?
→ Thank you for your comment. We designed the coordinated robotic hand with three fingers because we thought that is the minimal number of fingers to express coordinated movements of the human hand. The reason we chose the minimal number is to make a prototype quickly, and it is more cost-efficient. However, in principle, it is possible to increase the number of fingers of the robotic hand, if it is necessary to express more complex coordinated movement.

6. Please be careful when you use the word “spatiotemporal” because the input signals here are only about the effect from distance (space information). There is no effect of the time-related input signal that has been demonstrated in this manuscript.
→ Thank you for your comment. The term “spatiotemporal” was used because PSC was controllable with varying distance and frequency of the input signals. However, as the reviewer mentioned, we admit that our manuscript has little information to be referred as spatiotemporal effect and decided to change the term “spatiotemporal” to “spatial”.

7. Have authors considered the method of how to initiate the state of each finger after bending?
→ Thank you for the comment. The initiation of the robotic fingers could be performed by decreasing the PSC current. The PSC current can be decreased by natural decay or by applying LTD pulses to the gate. The initiation of the robotic fingers can be seen in Movie S1. At the releasing the cup, the grabbed fingers were returned to the original position by applied LTD pulses. For a better understanding of the initiation process of the robotic finger, we revised the manuscript as follows:

Line 219: In the case of the PPSA-based control system, the three robotic fingers successfully grabbed the cup firmly along the contour of cup through coordinated movement induced by the LTP pulses and released the cup when V_{IN} was decreased by the consecutive LTD pulses (**Movie S1**).

8. Authors also may consider cite more references in the context of electrochemical synaptic transistor's use for the robot manipulation in line 47-48, such as “A 3D-printed neuromorphic humanoid hand for grasping unknown objects”, Science, Volume 25, Issue 4, 15 April 2022, 104119.
→ Thank you for your valuable opinion. We added the citation as the reviewer recommended.

Line 47: Human synapses transmit biological signals by releasing neurotransmitters from presynaptic neurons; the neurotransmitters pass through the synaptic cleft and reach postsynaptic neurons^{18,20-22,28-30}

Reviewer: 2

The manuscript titled “Humanlike Spontaneous Motion Coordination of Robotic Fingers Through Spatiotemporal Multi-input Spike Signal Multiplexing” by Dong Gue Roe et al. exhibited a coordinated movement of the fingers was achieved with reduced control complexity by exploiting the advantages of parallel multiplexing and analog logic by the integration of the synaptic array and a robotic hand. In general, this work seems to be quite interesting and I support its publication but I have some comments..

1. The properties of ion gel dielectric can be influenced by external environment, such as humidity, temperature, thus the operating environment of robotic hand should introduce.
→Thank you for your thoughtful opinion. The robotic hand operation in this manuscript was conducted on ambient room temperature environment. As the reviewer mentioned we examined the operating condition of our devices as follows.

Fig S5. (a) Transfer curves under varying temperatures. (b) Transfer curves under varying relative humidity.

To investigate the effect off external environment, we examined the transfer characteristics of the transistors under varying temperatures and relative humidity. Along with increasing temperatures, the on/off ratio at 0 V was maintained at the level of 10^3 until 70 °C (Supplementary Fig. 5a). Also, we conducted the same experiment under the varying relative humidity (Supplementary Fig. 5b). Negligible transfer curve change was observed until humidity of 80 RH%. These results show that ion gel dielectric used in the device can endure temperature and humidity changes. The information above is added in the Supplementary Fig. 5 and we revised manuscript as follows:

Line 135: Also, repeatability and environmental stability of the OAST was evaluated by applying 8,000 LTP/D pulses and changing temperature and humidity, respectively (**Supplementary Fig. 5**).

2. How about the repeatability of transistor and the bending angle of robotic finger?
→Thank you for the comment. First, we tested repeatability of the transistor by applying 8,000 consecutive LTP/D pulses (Supplementary Fig 5c). Except for the slight increase in the peak PSC current, no significant changes occurred during the repeatability test. We also conducted the experiment to check the repeatability of the robotic finger's bending angle. The robotic finger system was actuated 15 times by connecting robotic finger to PPSA. As shown in Supplementary Fig. 13, the bending angle of the actuator presented excellent repeatability with negligible bending angle change.

Fig. S5. (c) Repeatability test results of the PPSA.

Fig S13. Repeatability test results of the robotic finger.

The information above is added in the Supplementary Fig. 5 and 13, and we revised manuscript as follows:

Line 135: Also, repeatability and environmental stability of the OAST was evaluated by applying 8,000 LTP/D pulses and changing temperature and humidity, respectively (**Supplementary Fig. 5**).

Line 189: The repeatability of the robotic finger was tested by performing 15 consecutive actuation tests (**Supplementary Fig. 13**).

3. How long the coordinated robotic actuation of grabbing an object with curvature can maintain?
 - Thank you for the comment. To test how long robotic hand can maintain the grabbing motion with same curvature, we have redone the cup grabbing experiment with longer time period. As shown in the **Revision video 1**, our coordinated robotic hand could easily maintain the cup grabbing motion for more than 9 minutes. Considering the durability of the global gate synaptic transistor as presented in Supplementary Fig. 5c, we believe coordinated robotic hand can maintain the grabbing motion for more than 9 minutes.
4. In fig2d, there's some overlap between 3 and 4.
 - Thank you for the opinion. For better readability, we reduced the size of the symbol as follows:

- 1 Koutsouras, D. A. et al. An Iontronic Multiplexer Based on Spatiotemporal Dynamics of Multiterminal Organic Electrochemical Transistors. *Adv. Funct. Mater.*, 2011013 (2021).

REVIEWERS' COMMENTS

Reviewer #3 (Remarks to the Author):

The questions raised by the author have been answered, and the article has been modified. The article puts forward its own idea on the realization of the humanlike spontaneous motion coordination of robotic fingers.

The author conducts an experimental discussion on this idea, and the article conception and related work are improved.